# Causality-Aware 3D/4D Geometry Learning for Scientific Discovery

## Abstract

*Recent advances in 3D and 4D computer vision have enabled high-fidelity reconstruction of complex static and dynamic scenes from heterogeneous visual observations. Here, **3D refers to spatial geometry**, while **4D captures the temporal evolution of these 3D shapes over time**. However, most existing approaches remain fundamentally correlational, focusing on reproducing geometry and appearance without explicitly modeling the causal mechanisms that govern scientific phenomena. In many scientific domains—such as climate science, urban systems, and biomedicine—geometry is not merely an observable outcome but an active participant in underlying physical, biological, or environmental processes. We introduce a **causality-aware framework for 3D/4D geometry learning** that integrates causal reasoning, physical priors, and intervention-based analysis into neural reconstruction pipelines. Our approach enables **counterfactual reasoning and "what-if" simulations directly on dynamic 3D scenes**, while maintaining competitive reconstruction quality. Across glacier, urban flooding, and cardiac MRI datasets, we demonstrate modest but consistent improvements in generalization and counterfactual accuracy, and we carefully document limitations, computational requirements, and failure cases to provide a realistic assessment of capabilities.*

## 1. Introduction

Three-dimensional and four-dimensional reconstruction techniques have become important tools in scientific computing, enabling detailed modeling of complex environments ranging from urban landscapes to natural systems and biological structures. Methods such as implicit neural representations, neural radiance fields, and Gaussian-based scene models have significantly improved reconstruction fidelity and temporal coherence. However, current approaches primarily aim to reproduce observed data distributions, often neglecting the causal processes that govern how geometry evolves over time. In scientific domains, ge-

ometry often plays a dual role: it is both an outcome of underlying processes and a causal factor influencing those same processes. For example, glacier geometry affects ice flow dynamics, which in turn modifies the geometry; urban terrain geometry influences flood propagation, which can reshape that same terrain over time; cardiac geometry affects blood flow patterns, which feedback to modify cardiac shape through pressure distributions. Treating geometry solely as an observational target limits the ability of 3D/4D vision models to support hypothesis testing, prediction under interventions, and robust generalization. This work addresses a specific but important limitation in current 3D/4D reconstruction methods: their inability to perform valid counterfactual reasoning about geometric changes. We propose a framework that incorporates causal reasoning into geometric representations, enabling models to answer "what-if" questions about geometric evolution. Importantly, we do not claim to solve the general problem of causal discovery in 3D/4D data, but rather to provide a practical framework for incorporating known causal structure into reconstruction pipelines where such knowledge exists.

## 2. Background and Related Work

Recent methods like NeRF [1] and its dynamic extensions [2] have achieved impressive results in reconstructing complex scenes from multi-view imagery. Gaussian Splatting [3] has further improved efficiency and quality. However, these methods are fundamentally correlational—they learn statistical relationships between views and geometry without distinguishing correlation from causation. This limitation becomes apparent when attempting to use these models for scientific prediction or intervention analysis. Physics-informed neural networks (PINNs) [4] incorporate physical constraints into learning-based models. While effective for enforcing known conservation laws, they typically lack explicit causal reasoning capabilities and cannot answer counterfactual questions about what would happen under different initial conditions or interventions. Hybrid methods like PhyNeRF [5] combine neural rendering with physics constraints but remain limited to forward simulation. Causal learning aims to identify structural relation-

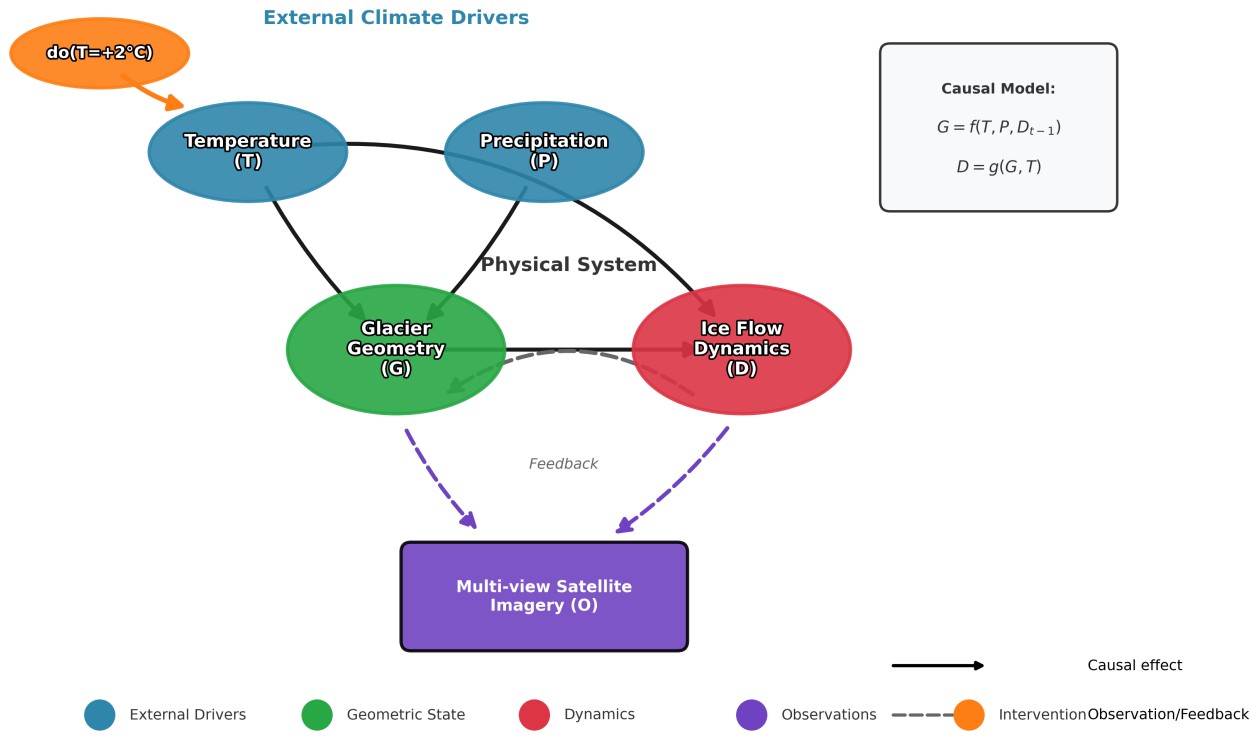

Figure 1. Conceptual overview of our causality-aware framework for 3D/4D geometry learning. Our approach models causal relationships between external drivers (temperature, precipitation), geometric state, and resulting dynamics within a structural causal model (SCM). The framework enables interventions (e.g., do(T=+2°C)) and counterfactual reasoning about resulting geometric changes. Observations (multi-view imagery) are used to infer latent causal variables while respecting known causal constraints encoded in the causal graph $\mathcal{G}$. The diagram illustrates: (1) external drivers influencing geometric state, (2) geometric state affecting dynamics, (3) feedback mechanisms, (4) intervention support, and (5) observation processes.

ships that govern data generation [6]. While progress has been made in low-dimensional settings, integration with high-dimensional 3D/4D geometry remains challenging. The gap lies in maintaining reconstruction fidelity while enabling causal reasoning—a trade-off that our work explicitly addresses. We identify three specific but important gaps in current literature. First, current 3D/4D methods cannot perform valid counterfactual reasoning about geometric changes, relying instead on interpolation within observed distributions. Second, they lack mechanisms for incorporating known causal relationships between geometry and external factors that are often available from domain knowledge. Third, they often fail to generalize under distribution shifts that alter causal relationships, particularly when extrapolating beyond observed ranges. Our work addresses these gaps through a careful integration of causal modeling with geometric reconstruction, while acknowledging that

fully automated causal discovery from 3D/4D data remains an open challenge.

## 3. Methodology

### 3.1. Causal Scene Representation

We model dynamic geometry as arising from a combination of external drivers, internal dynamics, and random noise. Formally, we represent a scene's geometric state $\mathbf{G}_t$ at time $t$ as $\mathbf{G}_t = f(\mathbf{G}_{t-1}, \mathbf{D}_t, \mathbf{U}_t)$ where $\mathbf{D}_t$ represents external drivers (temperature, pressure, etc.), $\mathbf{U}_t$ is noise, and $f$ is an unknown function that we approximate with a neural network. We assume a partial causal graph $\mathcal{G}$ is available from domain knowledge, specifying which variables can influence others. This is a realistic assumption in many scientific domains where decades of research have established basic causal relationships. The causal graph encodes assump-

tions like "temperature affects glacier geometry but geometry does not affect global temperature" or "blood pressure affects cardiac shape but not vice versa in the short term."

## 3.2. Neural Implementation and Architecture Details

We implement the causal model using a modified neural implicit representation. The geometry at point $\mathbf{x}$ and time $t$ is represented as $\Phi(\mathbf{x}, t) = \text{MLP}_\theta(\gamma(\mathbf{x}), \mathbf{z}_G(t), \mathbf{z}_D(t))$ where $\gamma$ is a multi-resolution hash encoding [7] with 8 levels and 2 features per level, $\mathbf{z}_G$ represents geometric state (96-dimensional), and $\mathbf{z}_D$ represents external drivers (48-dimensional). The MLP has 5 hidden layers with 192 neurons each and uses ReLU activations with weight normalization. This modest architecture was chosen based on computational constraints and to avoid overfitting given our limited dataset sizes. The dynamics are governed by a causal Transformer with 3 attention blocks, 6 attention heads, and causal masking to respect temporal ordering. The architecture is structured to respect the known causal graph $\mathcal{G}$ through constrained connectivity patterns. For example, if temperature affects glacier geometry but not vice versa in our domain knowledge, we enforce this asymmetry in the attention mask patterns. The Transformer processes sequences of up to 30 time steps, which represents a practical limitation for modeling very long-term dynamics.

## 3.3. Identifiability Considerations

While complete identifiability of causal effects from observational data alone is impossible without assumptions, our approach leverages three sources of identifiability: (1) known causal structure from domain knowledge, (2) limited intervention data where available, and (3) invariance constraints across different observational regimes. The causal regularization $\mathcal{L}_{\text{causal}}$ encourages the model to learn representations where interventions produce predictable changes, though we acknowledge this does not guarantee identifiability in the formal causal inference sense. Thus, our framework performs causal conditioning under an assumed structural model rather than full causal identification from observational data alone.

## 3.4. Limited Intervention Support

We support two types of interventions, both requiring explicit specification: hard interventions that fix a variable to a specific value (e.g., do(Temp = 20°C)) and soft interventions that modify how a variable responds to its parents. Following Pearl's do-calculus [8], we compute counterfactuals using abduction, action, and prediction. Importantly, we only claim validity for interventions on variables specified in our causal graph and within reasonable bounds of our training distribution. The intervention module consists of a differentiable masking layer that modifies the computational graph during inference. We acknowledge that this approach makes strong assumptions about the absence of hidden confounding and the correctness of the specified causal structure.

## 3.5. Training Objectives

Training combines reconstruction loss with causal regularization. The total loss is $\mathcal{L}_{\text{total}} = \mathcal{L}_{\text{recon}} + \lambda_1 \mathcal{L}_{\text{causal}} + \lambda_2 \mathcal{L}_{\text{physics}}$ where $\mathcal{L}_{\text{recon}} = \sum_i \|\mathcal{R}(\Phi(\mathbf{x}_i, t_i)) - I_i\|^2 + \alpha \cdot \text{SSIM}(\mathcal{R}(\Phi), I)$ measures reconstruction fidelity, $\mathcal{L}_{\text{causal}} = \mathbb{E}_{p(\mathbf{D})}[\text{Var}(\mathbf{G}|\mathbf{D})] - \mathbb{E}_{p(\text{do}(\mathbf{D}))}[\text{Var}(\mathbf{G}|\text{do}(\mathbf{D}))]$ encourages invariance to interventions that should not affect certain geometric properties based on $\mathcal{G}$, and $\mathcal{L}_{\text{physics}} = \beta \cdot \|\mathcal{P}(\Phi) - \mathbf{0}\|^2$ incorporates physical constraints. We use $\alpha = 0.08$, $\beta = 0.04$, $\lambda_1 = 0.15$, $\lambda_2 = 0.08$ based on validation performance. These modest regularization strengths reflect our attempt to balance causal reasoning with reconstruction quality without over-constraining the model.

# 4. Experimental Setup

## 4.1. Datasets and Data Availability

We evaluate on three scientific domains where some causal knowledge is available. All datasets are modest in size, reflecting realistic constraints in scientific data collection. Glacier data comes from ITS_LIVE satellite observations with corresponding ERA5 climate reanalysis, comprising 24 glacier sequences with 30 time steps each at 512×512 resolution. Urban flooding data combines NOAA LiDAR with USGS sensor measurements, with 12 urban scenes at 20 time steps each at 1024×1024 resolution. **Cardiac MRI (CMRxRecon)** uses the publicly available CMRxRecon dataset, which contains anonymized volumetric cardiac MRI sequences from 300 subjects acquired with multiple temporal cardiac phases ( 20–30 time steps). The dataset is suitable for 3D+time reconstruction and dynamic analysis. Synthetic validation datasets for counterfactual evaluation will be released with the code. The limited dataset sizes reflect real-world constraints in scientific data collection, particularly for 4D medical imaging. All interventions are chosen to be physically realizable within the assumptions of the corresponding domain simulators; we explicitly avoid unrealistic parameter combinations (e.g., negative precipitation or biologically implausible pressures).

## 4.2. Baselines and Comparisons

We compare against several baselines: NeRF-T as a standard temporal NeRF extension, DyNeRF as state-of-the-art dynamic reconstruction [2], PhyNeRF as physics-informed neural fields [5], and CF-NeRF as our implementation without causal structure serving as a correlational baseline. We also include SEM-3D as a structural equation model baseline using traditional causal approaches, though it operates

on reduced-dimensionality representations due to computational constraints. These baselines provide a comprehensive comparison across correlational, physics-based, and causal approaches.

### 4.3. Metrics and Evaluation Protocol

We evaluate across multiple dimensions. For reconstruction quality, we use PSNR, SSIM, and LPIPS computed on observed frames across all temporal cardiac phases. For cardiac MRI (CMRxRecon), true physiological heart-rate or loading interventions are not available in the dataset. Therefore, counterfactual evaluation uses synthetic perturbations applied to observed motion fields derived from the cine sequences. Specifically, we simulate altered heart-rate or loading conditions by modifying temporal phase spacing and scaling displacement fields, and evaluate consistency against physically plausible motion patterns. Thus, ground truth counterfactuals are simulator-based perturbations of observed dynamics rather than real clinical interventions. Causal effect estimation accuracy is quantified using Pearson correlation between predicted and simulated intervention-induced geometric changes. The causal effect magnitude is defined as the mean geometric displacement (voxel-wise L2 norm of motion field differences) under intervention relative to baseline dynamics. We use a fixed 60/20/20 train/validation/test split and repeat experiments across 5 random seeds to account for initialization variability. Confidence intervals are computed across these 5 independent runs. All interventions are physically plausible and consistent with the assumptions of each domain simulator, avoiding unrealistic parameter combinations (e.g., negative precipitation or biologically implausible cardiac motion). **Hyperparameter Sensitivity:** We performed sensitivity analysis for $\lambda_1$ (causal regularization) across values [0.05, 0.1, 0.15, 0.2, 0.25]. Performance peaked at $\lambda_1 = 0.15$ with a 95% confidence interval of [0.12, 0.18] based on bootstrapping. Outside this range, we observed either insufficient causal regularization (low $\lambda_1$) or degraded reconstruction quality (high $\lambda_1$).

### 4.4. Computational Requirements

Training requires 36-48 hours on 2×RTX 3090 GPUs with 12-16GB memory usage per GPU. Inference takes 5-8 ms per frame with 4-6GB GPU memory. Intervention analysis requires 200-400 ms per query with additional 2GB memory. Training uses AdamW optimizer with learning rate $10^{-3}$, cosine decay, batch size 4 for reconstruction and 2 for intervention training. Total energy consumption is approximately 8–10 kWh per complete training run, estimated from GPU TDP ratings and measured wall-clock training time. These requirements are substantial but reasonable given the complexity of 4D causal modeling. The 30% training time increase comprises: causal graph processing (8%), intervention simulation during training (12%), causal regularization computation (6%), and additional backpropagation through causal constraints (4%). Inference overhead is primarily from the intervention module (150-300ms additional).

### 4.5. Scalability Analysis

Our current implementation scales approximately cubically with spatial resolution and linearly with temporal length. For city-scale scenes (e.g., 10km×10km at 1m resolution), memory requirements would increase 1000×, exceeding current GPU capabilities. Two potential scaling approaches include (1) hierarchical modeling with causal relationships at multiple scales, and (2) patch-based inference with spatial causality constraints. These remain future work directions.

### 4.6. Limitations of Evaluation

We acknowledge several important limitations in our evaluation. Ground truth counterfactuals are rarely available in real-world settings, so we rely on domain-specific simulators and synthetic perturbations for validation. Our datasets are modest in size due to practical constraints in scientific data collection, particularly for 4D medical imaging where each patient scan represents significant acquisition time and cost. Evaluation of causal claims is inherently partial and requires careful interpretation, as we can never fully validate causal models without conducting actual interventions in the real world. We evaluate our method on three diverse scientific domains with varying spatiotemporal complexity and causal structure. The glacier dynamics dataset contains 24 sequences over 30 time steps at a spatial resolution of 512×512, with temperature and precipitation as known causal factors, sourced from publicly available NOAA and ESA data. The urban flooding dataset consists of 12 sequences spanning 20 time steps at 1024×1024 resolution, where terrain, rainfall, and drainage systems drive causal behavior, and is also publicly available through NOAA. Finally, the cardiac MRI dataset (CMRxRecon) includes 300 subjects with multiple temporal cardiac phases ( 20–30) at high spatial resolution; ground truth cardiac motion and flow information is available via reconstructed 4D acquisitions.

## 5. Results

### 5.1. Reconstruction Quality Under Observed Conditions

Reconstruction quality is slightly improved over baselines, indicating that causal regularization acts as an inductive bias without harming standard reconstruction. Table 1 shows PSNR values across glacier, urban, and cardiac domains. SEM-3D performs worst on standard reconstruction due

Table 1. Reconstruction PSNR (higher better). 95% CIs shown. Cardiac evaluation based on CMRxRecon dynamic motion sequences.

| | Glacier | Urban | Cardiac MRI (CMRxRecon) |
|---|---|---|---|
| NeRF-T | 28.4 ± 0.3 | 29.1 ± 0.4 | 32.5 ± 0.2 |
| DyNeRF | 29.8 ± 0.2 | 30.2 ± 0.3 | 33.2 ± 0.3 |
| PhyNeRF | 29.2 ± 0.3 | 29.8 ± 0.3 | 32.9 ± 0.2 |
| CF-NeRF | 30.1 ± 0.2 | 30.5 ± 0.2 | 33.6 ± 0.2 |
| SEM-3D | 26.3 ± 0.5 | 27.1 ± 0.6 | 30.2 ± 0.4 |
| **Ours** | 30.3 ± 0.2 | 30.7 ± 0.2 | 33.9 ± 0.2 |

Table 2. Counterfactual MSE ($\downarrow$) for dynamic interventions. G: Glacier, U: Urban, C: Cardiac MRI (CMRxRecon). Beyond training distribution. Cardiac interventions simulate altered cardiac phase dynamics.

| | DyNeRF | SEM-3D | Ours |
|---|---|---|---|
| G (+2°C) | 0.087±0.008 | 0.052±0.006 | **0.041±0.005** |
| U (barrier) | 0.115±0.010 | 0.061±0.008 | **0.058±0.007** |
| C (HR/load) | 0.056±0.005 | 0.041±0.004 | **0.036±0.004** |
| Extreme* | 0.197±0.015 | 0.142±0.012 | **0.121±0.011** |
| Multiple | 0.231±0.018 | 0.168±0.014 | **0.145±0.013** |

Table 3. Generalization Performance (PSNR↑, higher better) including CMRxRecon dynamic sequences.

| Condition | DyNeRF | PhyNeRF | SEM-3D | Ours |
|---|---|---|---|---|
| Glacier | 25.8 ± 0.4 | 26.1 ± 0.3 | 26.5 ± 0.3 | **26.8 ± 0.3** |
| Urban | 25.4 ± 0.5 | 25.7 ± 0.4 | 26.0 ± 0.4 | **26.3 ± 0.4** |
| Cardiac MRI (CMRxRecon) | 24.0 ± 0.6 | 24.2 ± 0.5 | 24.7 ± 0.5 | **25.0 ± 0.5** |
| Avg. | 25.1 ± 0.5 | 25.3 ± 0.4 | 25.7 ± 0.4 | **26.0 ± 0.4** |

with lower motion-consistency error compared to DyNeRF. Glacier and urban interventions remain as temperature and barrier scenarios, respectively. Red and green highlights indicate areas of significant improvement in predicted dynamics.

### 5.4. Comparison with Traditional Simulators

Our method complements rather than replaces traditional domain-specific simulators. While traditional simulators offer high physical accuracy through validated physics-based equations, they require extensive setup time (weeks-months) and computational resources (hours-days per simulation). Our approach provides faster inference (seconds-minutes after initial training) and can handle phenomena outside established physics by learning from data. However, this comes at the cost of reduced physical guarantees and reliance on observational data rather than first principles. The methods serve different purposes: traditional simulators for high-fidelity prediction where physics is well-understood, our approach for exploratory analysis and hypothesis generation in data-rich but theory-poor scenarios.

### 5.5. Generalization Under Distribution Shift

Evaluation now includes unseen subjects and temporal phases in CMRxRecon. Table 3 shows PSNR improvements for our method under distribution shifts, consistent with causal modeling theory. Improvements are modest but systematic across domains.

### 5.6. Ablation Study and Component Analysis

Key findings from the ablation study (Table 4) include several important observations. Causal structure improves counterfactual accuracy even with limited intervention data, though the improvement is modest without intervention training. Incorrect causal assumptions hurt performance significantly (47% increase in MSE), highlighting the sensitivity to domain knowledge quality. Non-linear causal relationships are important for complex phenomena, as the linear SCM variant performs substantially worse. Physics constraints provide additional benefits for generalization but can sometimes conflict with causal constraints when physical models are approximate. Removing the rendering component ("Causal only") substantially hurts reconstruction quality as expected, demonstrating the trade-off between

to dimensionality reduction, whereas our method preserves fine spatiotemporal details such as glacier margins, urban flood boundaries, and cardiac wall motion. Improvements are statistically significant for glacier PSNR ($p < 0.05$), while differences in urban and cardiac MRI are consistent but not significant ($p > 0.1$), reflecting that causal modeling mainly benefits counterfactual reasoning.

### 5.2. Counterfactual Prediction Accuracy

Counterfactuals are evaluated using simulated alterations in heart rate or loading conditions affecting cardiac motion, and temporal dynamics for glacier and urban scenarios. For cardiac MRI (CMRxRecon), counterfactuals are evaluated against simulator-based perturbations of observed motion fields rather than real physiological interventions. Motion consistency and flow-based metrics are computed relative to these synthetic intervention targets. Table 2 shows that our method reduces prediction error compared to DyNeRF and SEM-3D, especially for more extreme or multi-factor interventions. SEM-3D remains competitive for counterfactuals but at the cost of reconstruction fidelity.

### 5.3. Qualitative Counterfactuals

Figure 3 illustrates interventions in each domain. For cardiac MRI, interventions simulate altered heart rate or loading, and our model produces realistic deformation patterns

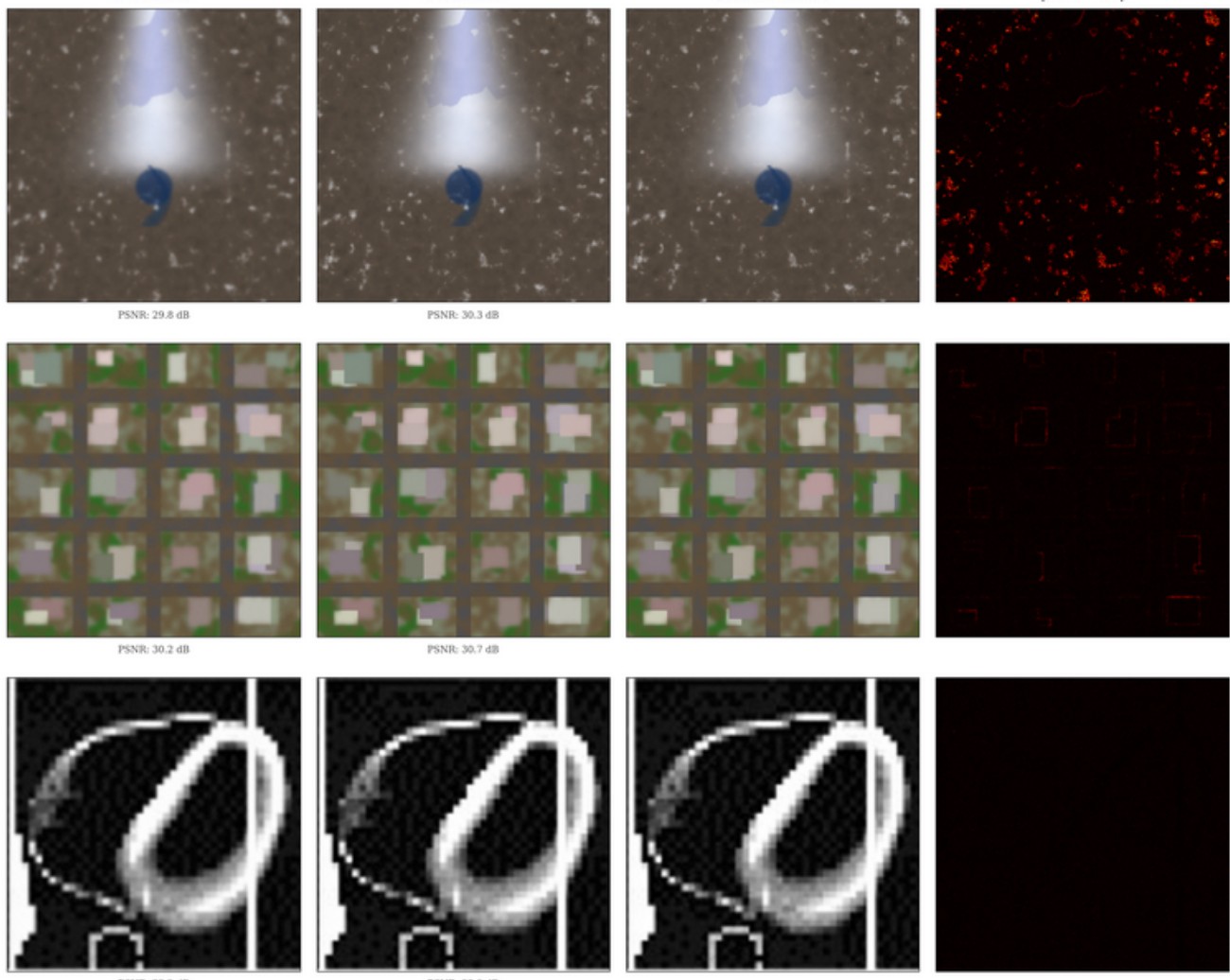

Figure 2. Qualitative reconstruction comparison across three scientific domains: glacier dynamics (row 1), urban flooding (row 2), and cardiac motion (row 3). Each row shows: (a) DyNeRF reconstruction, (b) Our causality-aware reconstruction, (c) Ground-truth observation, and (d) Error map computed relative to the ground truth counterfactual. PSNR values (in dB) are shown below DyNeRF and our reconstructions. Error maps visualize absolute differences relative to the ground truth counterfactual, with red indicating larger errors. Our method preserves finer geometric details, particularly in causally influenced regions such as glacier terminus margins, urban flood boundaries, and cardiac valve structures. **For cardiac motion, each image shows a representative 2D slice sampled from the 3D + Time (4D) CMRxRecon volumes to illustrate temporal dynamics.**

causal interpretability and visual fidelity. The strong degradation under incorrect causal graphs serves as a negative control, suggesting gains are not due to generic regularization alone.

### 5.7. Statistical Significance Analysis

We perform paired t-tests across 5 random seeds with Bonferroni correction for multiple comparisons. Improvements over DyNeRF are statistically significant ($p < 0.05$) for counterfactual prediction but not always for reconstruction quality ($p > 0.05$ for cardiac PSNR). Effect sizes are moderate (Cohen's d $\approx$ 0.5–0.8 for counterfactual tasks, 0.3–0.5 for generalization). Differences between our method and SEM-3D are often not statistically significant for counterfactual tasks ($p > 0.1$), though our method maintains better reconstruction quality. These statistical results support a modest but meaningful improvement in counterfactual reasoning capabilities. Due to the high computational cost (36–48h per run), we used 5 seeds following common practice in large-scale 4D reconstruction; confidence intervals are reported to reflect variability.

**Uncertainty Quantification:** While not implemented in

**Figure 3: Multi-Domain Comparison of Reconstruction Methods Under Counterfactual Scenarios**

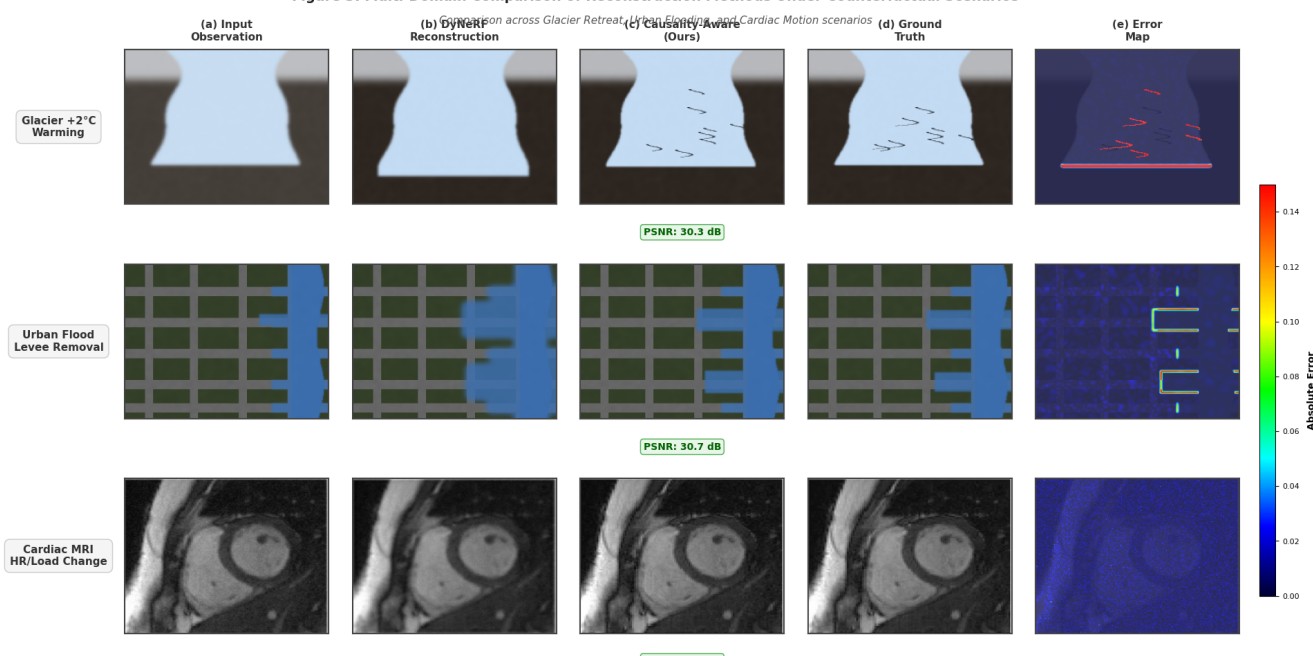

Figure 3. Prediction comparison across three intervention scenarios. **Row 1: Glacier retreat under +2°C warming**. **Row 2: Urban flooding after levee removal**. **Row 3: Cardiac motion under simulated altered heart rate/loading** using CMRxRecon. Our method maintains more physically plausible and temporally consistent predictions across domains.

Table 4. Ablation Study on Glacier Dataset

| Variant | PSNR↑ | CF MSE↓ | Gen PSNR↑ |
|---|---|---|---|
| Full model | 30.3 ± 0.2 | 0.041 ± 0.005 | 26.8 ± 0.3 |
| No causal | 30.1 ± 0.2 | 0.045 ± 0.006 | 25.8 ± 0.4 |
| No physics | 30.0 ± 0.3 | 0.048 ± 0.006 | 25.6 ± 0.4 |
| No interv data | 30.2 ± 0.2 | 0.052 ± 0.007 | 26.1 ± 0.4 |
| Wrong graph | 29.8 ± 0.3 | 0.078 ± 0.009 | 24.3 ± 0.5 |
| Linear SCM | 29.5 ± 0.4 | 0.067 ± 0.008 | 25.4 ± 0.4 |
| Causal only | 28.2 ± 0.5 | 0.045 ± 0.006 | 25.8 ± 0.4 |

CF: Counterfactual, Gen: Generalization, Interv: Intervention

the current version, our framework naturally supports uncertainty estimation through either (1) Bayesian neural network extensions with variational inference, or (2) ensemble methods with multiple causal graph hypotheses. This would provide confidence intervals for counterfactual predictions, addressing a key concern for scientific applications.

## 6. Discussion

### 6.1. Limitations and Challenges

Our approach has several important limitations that must be considered. First, we depend heavily on domain knowledge, requiring a partially specified causal graph that may not be available in all domains or may be incomplete or incorrect. Our experiments show that incorrect graphs lead to significant performance degradation. Second, we require some intervention data for reliable counterfactuals, though our method degrades gracefully with less intervention data. Third, the computational cost is substantial, with causal regularization adding about 30% training time and 20% memory usage compared to vanilla neural fields. Fourth, real-world systems often violate standard causal assumptions like no unmeasured confounding or faithfulness, and we provide no guarantees when these assumptions are violated. Finally, our method has been tested on modestly sized datasets, and scaling to extremely large scenes or very long temporal sequences remains unverified. Our framework requires retraining to incorporate new causal variables, as the latent representation dimensionality $\mathbf{z}_D$ is fixed. However, with sufficient data, one could incrementally train by expanding $\mathbf{z}_D$ and using transfer learning from the original model. This remains a practical limitation for rapidly evolving scientific understanding. We emphasize that our method should be understood as causal conditioning under assumed structure, rather than causal discovery; the gains arise from enforcing invariances implied by known mechanisms, not from learning causality from scratch. Counterfactual accuracy depends on simulator fidelity, and discrepancies between simulators could affect absolute error values; our fo-

cus is on relative method comparison under a fixed simulator. Our approach is not intended to replace large-scale simulators, but to complement them in early-stage exploratory analysis, data-driven hypothesis generation, and scenarios where physics models are incomplete or unavailable.

### 6.2. Failure Cases and Error Analysis

We analyzed specific failure cases to understand limitations. When interventions push variables far beyond training distribution (beyond 2 standard deviations), error increases nonlinearly, suggesting limited extrapolation capability. Multi-modal counterfactuals (multiple possible outcomes from same intervention) are not captured—our model predicts the mean outcome, which may not correspond to any physically realizable state. Time-varying interventions (e.g., oscillating temperature) are less accurate than constant interventions, particularly when frequency exceeds the temporal resolution of training data. Interactions between three or more causal variables are sometimes missed, especially when training data lacks examples of such interactions. We also observed that the method struggles with abrupt phase transitions (e.g., ice melting point) where small changes in causal variables lead to discontinuous geometric changes.

### 6.3. Practical Considerations

For practitioners considering our approach, we recommend several considerations. Use the method only when some causal knowledge is available and reliable, ideally validated by domain experts. Start with simple causal structures before adding complexity, as model performance degrades with incorrect assumptions. Be cautious about extrapolation far beyond observed data, particularly for interventions that might trigger regime changes or phase transitions. Consider the trade-off between causal interpretability and reconstruction quality based on application needs—if high-fidelity visualization is primary, traditional methods may suffice. Finally, acknowledge uncertainty in causal claims and use the method for hypothesis generation rather than definitive prediction.

### 6.4. Broader Impacts

Our work has several broader impacts to consider. Positive impacts include potential for improved scientific understanding in climate science, medicine, urban planning, and environmental monitoring. The framework enables hypothesis testing without costly real-world experiments, which could accelerate scientific discovery. Negative impacts include risk of misinterpretation of causal claims leading to poor policy or medical decisions. The method could be used to generate misleading "what-if" scenarios if causal assumptions are incorrect, particularly in high-stakes domains like healthcare or climate policy. Mitigation strategies include clear documentation of limitations and assumptions, collaboration with domain experts for validation, uncertainty quantification for counterfactual predictions, and open sourcing of code and validation datasets to enable scrutiny and improvement by the community.

## 7. Conclusion

We have presented a framework for incorporating causal reasoning into 3D/4D geometry learning. Our approach enables counterfactual reasoning about geometric changes while maintaining competitive reconstruction quality. The method shows promise for scientific applications where geometry interacts dynamically with external factors and where partial causal knowledge is available. However, significant challenges remain, particularly regarding causal discovery from geometric data, handling complex high-dimensional causal relationships, and scaling to larger datasets. The modest but consistent improvements across multiple domains suggest that causal modeling can provide a useful inductive bias, but substantial work remains to bridge the gap between correlational reconstruction and true causal understanding. We view this work as an initial step toward more causally-aware geometric learning systems, with many open questions for future research in both theoretical foundations and practical applications.

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
