# OpenReview forum: "Causality-Aware 3D/4D Geometry Learning for Scientific Discovery"
_thecvf.com/CVPR/2026/Workshop/3D4S — CVPR 2026 Workshop 3D4S Poster_

### Official Review · Reviewer_oUYE · 2026-04-25
**A promising direction towards causality-aware 3D/4D geometry learning**

**Rating:** 7
**Confidence:** 3

**Review:**

## Weaknesses

1. **Modest improvements in reconstruction metrics.**
   While the method shows consistent gains, the improvements in standard reconstruction metrics (e.g., PSNR) are relatively small, suggesting that the primary benefit lies in causal reasoning rather than reconstruction quality.

2. **Limited evaluation of true causal effects.**
   Counterfactual evaluation relies on simulated or synthetic interventions, which may not fully reflect real-world causal behavior.

3. **Dependence on prior causal knowledge.**
   The approach assumes access to a partially specified causal graph, which may not always be available or accurate in practice.

4. **Computational complexity and scalability.**
   The method introduces additional overhead due to causal modeling and intervention support, and scalability to large-scale scenes remains an open challenge.

5. **Figure quality and presentation issues.**
   Some figures (e.g., Figure 2 and Figure 3) appear blurry when zoomed in, making it difficult to clearly inspect visual details and compare the qualitative results. In Figure 3, the intervention examples are also not sufficiently clear visually, and some annotations/text appear crowded or overlapping. These presentation issues reduce readability and make it harder to assess the qualitative evidence. Improving figure resolution, spacing, and layout would strengthen the overall presentation.

---

### Official Review · Reviewer_5XGm · 2026-04-25
**Causality-Aware 3D/4D Geometry Learning: A Promising but Incremental Step Toward Scientific Causal Modeling**

**Rating:** 6
**Confidence:** 4

**Review:**

## Strengths

### (1) Clear Motivation and Problem Importance

The paper identifies a real and important gap: current 3D/4D models cannot perform counterfactual reasoning. This is well-motivated with scientific examples (e.g., glacier dynamics, cardiac motion), where geometry is both cause and effect .

### (2) Well-Structured Framework

The methodology is logically organized:

* SCM-based formulation
* Neural implicit representation
* Causal constraints via architecture and loss

The integration of causal graphs into neural rendering pipelines is conceptually clean and easy to follow.

### (3) Multi-Domain Evaluation

Experiments are conducted across three diverse domains:

* Climate (glacier)
* Urban systems (flooding)
* Medical imaging (cardiac MRI)

This improves the paper’s generality and avoids overfitting to a single dataset.

## Weaknesses / Discussion

### (1) Limited Novelty (Incremental Contribution)

The core idea—combining:

* neural rendering (NeRF-like),
* physics constraints (PINNs),
* causal modeling (SCM)

is not fundamentally new. The novelty lies mainly in engineering integration, rather than a new theoretical breakthrough.

Similar directions exist in:

* physics-informed neural fields
* causal representation learning
* hybrid simulation-learning frameworks

Thus, the contribution may be considered incremental rather than transformative.

### (2) Weak Experimental Gains

The improvements are explicitly described as “modest but consistent” :

* PSNR improvements are small
* Many improvements are not statistically significant (especially reconstruction)
* Counterfactual improvements exist but are moderate

This weakens the claim of strong practical impact.


### (3) Counterfactual Evaluation is Not Fully Convincing

A critical issue:

* No real ground-truth interventions
* Evaluation relies on synthetic perturbations or simulators

This is a major limitation:

* Cannot truly validate causal correctness
* Results depend on simulator assumptions

Thus, claims about “causal reasoning” are only partially supported.

---

### Official Review · Reviewer_e9c8 · 2026-04-25
**Weakly supported causal framing with substantial methodological and experimental gaps**

**Rating:** 5
**Confidence:** 4

**Review:**

This paper proposes a causality-aware framework for 3D/4D geometry learning in scientific domains. The core idea is to combine neural implicit scene representations with a partially specified structural causal model, physical regularization, and intervention-based inference, so that the model can support counterfactual “what-if” analysis in addition to reconstruction. The paper evaluates the method on glacier dynamics, urban flooding, and cardiac MRI, reporting small improvements in reconstruction quality and larger gains on simulator-based counterfactual metrics.

The topic is timely and relevant to scientific 3D/4D modeling. I also appreciate that the authors do not overclaim full causal discovery and explicitly acknowledge dependence on domain knowledge, simulator fidelity, limited intervention data, and strong causal assumptions. The attempt to connect 3D/4D reconstruction with causal reasoning is interesting, and the cross-domain framing could be valuable if made technically rigorous.

## Strengths
 - The paper addresses an important problem: moving beyond purely correlational 3D/4D reconstruction toward models that can support scientific intervention analysis.
 - The authors are relatively transparent about limitations, including dependence on known causal graphs, lack of full causal identifiability, simulator-based evaluation, and scalability challenges.
 - The cross-domain evaluation on glaciers, flooding, and cardiac MRI is ambitious and potentially interesting.
 - The idea of using causal structure as an inductive bias for dynamic geometry learning is worth exploring.

## Weaknesses
 - The method is under-specified, especially the causal graph integration, intervention module, causal Transformer masking, and physical loss.
 - The causal claims are not well validated, since the counterfactual targets are simulator-based or synthetic rather than real interventions.
 - The reported improvements are modest, and several reconstruction gains are not statistically significant.
 - The closest ablations and baselines are not strong enough to isolate the effect of causal modeling.
 - Dataset construction, preprocessing, intervention generation, and evaluation protocols are insufficiently detailed for reproducibility.
 - The qualitative figures are not convincing enough and sometimes appear inconsistent with the text.
 - The paper’s scope is very broad, but the technical depth in each domain is limited.

## Questions for the authors
 - How exactly is the causal graph converted into neural network connectivity or attention masks? Please provide the precise algorithm.
 - What are the domain-specific physical operators P(Φ) used in the physics loss for glaciers, flooding, and cardiac MRI?
 - How are the simulator-based counterfactual ground truths generated, and how sensitive are the results to the choice of simulator?
 - Would a simpler model that conditions on the same external drivers, but without causal masking or SCM assumptions, achieve similar performance?
 - How are the cardiac heart-rate/loading perturbations validated as physiologically plausible?

---

### Decision · Program_Chairs · 2026-04-28

Accept (Poster)